# Lipid Nanoparticles as Platforms for Theranostic Purposes: Recent Advances in the Field

Nikolaos Naziris [1,2,*] and Costas Demetzos [2]

1 Department of General Biophysics, Faculty of Biology and Environmental Protection, University of Lodz, Pomorska 141/143, 90-236 Lodz, Poland

2 Section of Pharmaceutical Technology, Department of Pharmacy, School of Health Sciences, National and Kapodistrian University of Athens, Panepistimioupolis Zografou, 15771 Athens, Greece; demetzos@pharm.uoa.gr

* Correspondence: nikolaos.naziris@biol.uni.lodz.pl; Tel.: +48-(42)-6354478

**Abstract:** Lipid nanoparticles (LNPs) are the first approved nanomedicines and the most well-studied class of nanocarriers for drug delivery. Currently, they are in the frontline of the pandemic fight as vaccine formulations and therapeutic products. However, even though they are so well-studied, new materials and new modifications arise every day that can improve their properties. Their dynamic nature, especially the liquid crystal state of membranes, is under constant investigation and it is that which many times leads to their complex biological behavior. In addition, newly discovered biomaterials and nanoparticles that possess promising effects and functionalities, but also toxicity and/or poor pharmacokinetics, can be combined with LNPs to ameliorate their properties. As a result, many promising theranostic applications have emerged during the past decade, proving the huge potential of LNPs in the field. In the present review, we summarize some of the most prominent classes of LNPs for nanotheranostic purposes, and present state-of-the-art research examples, with emphasis on the utilized biomaterials and the functionality that they confer to the resultant supramolecular nanosystems, in relation to diagnostic and therapeutic modalities. Although there has been unprecedented progress in theranostics, the translational gap between the bench and the clinic is undeniable. This issue must be addressed by experts in a coordinated way, in order to fully exploit these nanomedicines for the benefit of the society.

**Keywords:** lipid nanoparticles; liposomes; biomaterials; hybrid nanosystems; functional; theranostics; nanotheranostics; cancer; clinical translation

## 1. Introduction

Disease diagnosis and treatment have advanced greatly during recent decades. New therapeutic molecules and delivery systems are constantly being discovered, improving the effectiveness and safety of treatments, but also offering further benefits, such as the administration of already established medicines through alternative, more desirable routes, e.g., oral. In parallel, novel diagnostic tools are intensively investigated, in order to facilitate early detection of pathological states, which will lead to their timely treatment. However, the two fields need to be combined and applied synergistically, especially in the case of complex diseases, in order to ensure their maximum efficiency [1].

Theranostics is the unification between the ensembles of therapy and diagnostics. It is a hybrid emerging field that utilizes imaging techniques to supervise, in real-time, the efficacy and safety of therapies that target complex diseases, such as cancer [2]. It is clear that the monitoring modality can enhance the therapeutic efficacy of certain strategies, but there is more to it than meets the eye. It also facilitates the concept of personalized therapy, through in vivo visualization and real-time monitoring, thus ensuring the right dosage at the right time, depending on the individual and providing maximum efficiency with minimized risks.

Nanotheranostics is the application of the principles of nanotechnology in theranostics. It employs nanoparticles (NPs) or nanosystems (NSs) that integrate multiple functionalities on a single platform, truly broadening the possibilities of the field. These functionalities include the delivery of drugs, hence the term drug delivery systems (DDSs), as well as the delivery of molecules that serve as agents for imaging and diagnostic techniques. Computed tomography (CT), ultrasounds (US), magnetic resonance imaging (MRI), positron emission tomography (PET) and single photon emission computed tomography (SPECT) are some of the available diagnostic tools, while therapeutic strategies usually orientate towards chemotherapy, photodynamic therapy (PDT), photothermal therapy (PTT), ultrasound therapy (UT) and sonodynamic therapy (ST) [3]. Nanotheranostics is considered the next step in precision and personalized medicine, offering solutions to challenges that concern precise diagnosis, rational management and effective treatment of complex diseases, such as cancer [4].

Very important is the role of biomaterials in theranostics. Biomaterials are chemically synthesized, naturally occurring or semi-synthetic materials that are intended for application in biological organisms and comprise innovative excipients for novel pharmaceuticals and medicinal products. A biomaterial can, alone or as part of a complex system, direct a diagnostic or therapeutic effect through interactions with living systems. The biomaterial nature of a nanoparticle defines its class and properties. Lipid-based, polymer-based, protein-based, DNA-based, nucleoside-based, aptamer-based, inorganic and viral nanoparticles, as well as dendrimers, metal–organic frameworks (MOFs) and polymer-drug conjugates [4–6]. Additionally, regarding the intended application of nanotheranostics, they can offer solutions to a wide range of diseases, including cancer, pulmonary diseases, cardiovascular diseases, neurological disorders, etc. [5].

The aim of the present review is to summarize and present some of the latest advanced available technologies that integrate lipid-based nanoparticle (LNP) platforms and various types of functional materials, for concurrent diagnosis and treatment of diseases. The rapid advancements in the field over the past years, as well as those to come, necessitate frequent updates on the developed advanced technologies.

## 2. Lipid Nanoparticles and Mixed Nanosystems

LNPs are one major category of nanocarriers that are utilized as delivery systems of therapeutic and diagnostic molecules. Their main advantages over other classes include biocompatibility, biodegradability, safety, as well as incorporation and codelivery of both hydrophilic and hydrophobic molecules, with the latter enabling the theranostic concept and addressing the issue of real-time interaction between diagnosis and therapy. Liposomes, niosomes, transfersomes, nanoemulsions, solid lipid nanoparticles (SLNs), nanostructured lipid carriers (NLCs), lipid nanocapsules (LNCs), core–shell lipid nanoparticles (CLNs), lipid-based micelles and hybrid systems are representatives of this class of nanoparticles (Figure 1) [7,8]. These nanosystems are considered bioinspired because they mimic natural and physiological components and hold great promise as delivery vehicles for nanotheranostic applications [9].

The main ingredients of LNPs are lipids and surfactants, including phospholipids, cholesterol, solid and liquid lipids (fats and oils), as well as non-ionic and other surfactants [10]. In many of these cases, the components self-assemble into the liquid crystalline phase, which presents complex dynamics and metastable phases that are critical to the functionality of the systems [11,12]. LNPs can also contain lipids and materials with distinct structures and physicochemical characteristics, which lead to specific self-assembly, properties and biomedical applications (Table 1). The most recent well-known example is that of ionizable cationic lipids. These lipids have a $pK_a$ below 6.5 and during their development, they are charged, while they are non-charged in physiological pH. This class of lipids has been fully exploited for the development of some the recent COVID-19 vaccines. When mixed with genetic material, such as siRNA or mRNA, they produce complexes with internal hydrophobic and hydrophilic compartments. The genetic material resides in the

hydrophilic compartments; however, the morphology of these complexes varies, depending on the utilized lipid mixture (contains also a non-ionizable "helper" lipid), the N/P ratio (lipid amine to nucleotide phosphate) and the mixing process [13]. At the moment, the available technologies for COVID-19 vaccines include viral vectors, RNA-based vaccines and nano-adjuvants. The mRNA vaccines employ patented ionizable cationic lipids for the complexation of the genetic material, helper phospholipids, cholesterol, as well as a PEGylated surfactant or lipid that provides stealth properties and biological stability. The nano-adjuvant platform utilizes a recombinant protein-based nanoparticle that contains the virus spike protein, and an immune stimulating complex (ISCOM) matrix that is built by *Quillaja* saponins, cholesterol and phospholipids and acts as an adjuvant [13].

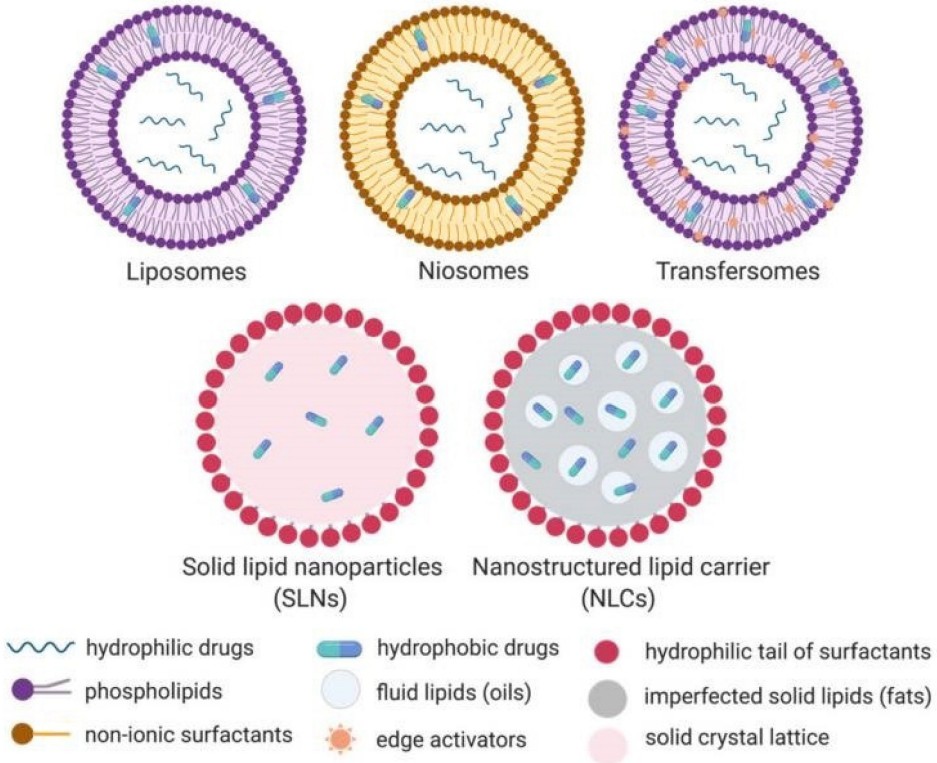

**Figure 1.** Different classes of lipid nanoparticles (LNPs). Reproduced from [10].

LNPs offer great versatility in technological platform design, through combination with other classes, leading to the creation of advanced mixed/chimeric nanosystems [14]. Additionally, these advanced nanosystems can also possess functional abilities that allow them to respond to physiological or externally applied stimuli, giving rise to the so-called stimuli-responsive or stimuli-sensitive nanosystems. Some of these stimuli include heat, pH alterations, magnetic field, ultrasounds and light, which after signal reception from the nanosystem, lead to structural rearrangement. This translates to alteration of its properties, e.g., size and membrane integrity, leading to spatiotemporal release of the incorporated or encapsulated contents [15].

LNPs are already established innovative excipients of nanomedicinal products and clinically established for many therapeutic purposes [16,17]. Therefore, it is rational to modify them through incorporation or conjugation of other functional components, with the purpose of rendering them suitable for concurrent diagnosis [8,18]. As theranostic tools, LNPs can improve the pharmacokinetics, the efficacy and safety of both the therapeutic and the imaging agents, while circumventing biological obstacles and targeting specific cells and tissues, through surface functionalization with targeting moieties [19]. Their advantages also include simple and up-scalable engineering processes, versatility through integration with other types of biomaterials, absence of toxicity, high degree of incorporation of both

hydrophilic and hydrophobic molecules and enhanced bioavailability [8]. Finally, from a regulatory point of view of the safety profile of LNPs, many of their lipid components are generally regarded as safe (GRAS), since they are present in the human body, as well as in food products that we consume [20].

**Table 1.** Commonly utilized headgroup-modified functionalized phospholipids.

| Molecule | Abbreviation | Application |
|---|---|---|
| * X-[methoxy(polyethylene glycol)-5000] (ammonium salt) | X-PEG5000 | Stealth® nanoparticles |
| X-[maleimide(polyethylene glycol)-2000] (ammonium salt) | X-PEG2000-MAL | Maleimide-functionalized thiol-reactive lipid for conjugation |
| X-(hexanoylamine) | - | Conjugation of triphenylphosphonium for mitochondrial targeting |
| X-(2,4-dinitrophenyl) (ammonium salt) | X-DNP | Antigenic nanoparticles |
| N-(4-carboxybenzyl)-N,N-dimethyl-2,3-bis(oleoyloxy)propan-1-aminium | DOBAQ | pH-sensitive nanoparticles |
| 1,2-dipalmitoyl-3-dimethylammonium-propane | DAP | Ionizable phospholipid for complexation of genetic material |
| X-diethylenetriaminepentaacetic acid (gadolinium salt) | X-DTPA (Gd) | MRI Imaging |
| X-(carboxyfluorescein) (ammonium salt) | X-CF | Fluorescent nanoparticles |
| X-(lissamine rhodamine B sulfonyl) (ammonium salt) | X-Liss Rhod | Fluorescent nanoparticles |

* X stands for phospholipid.

Probably the most prominent class of LNPs is liposomes, for which there are already numerous clinically approved formulations for various conditions. Liposomes are sphere-like nanostructures that consist primarily of phospholipids [21]. Their amphiphilic nature, composed of a lipid membrane surrounding an aqueous core, enables the codelivery of hydrophilic and hydrophobic or lipophilic molecules [22]. At the same time, their surface can be functionalized very easily for active targeting, stimuli-responsiveness or other purposes. This can be achieved either by physical incorporation of molecules inside the membrane during the self-assembly process, or by chemical modification of lipids and conjugation of functional moieties [23]. The final ensemble can exhibit different levels of complexity, from the inner core to the outer surface, which renders necessary the analysis of liposomes with various advanced tools.

Liposomes offer several important advantages as drug delivery systems, such as platform versatility, high loading of various types of diagnostic and therapeutic molecules, high biological stability, tunable drug release kinetics and biocompatibility [24]. In addition, liposomes are one of the most well-studies DDSs, with knowledge from the 1960s and the first liposomal formulation being approved by the FDA as the first nano-drug in 1995 [25]. Their production methods are very well-known and well-tuned, even in industrial scale. What is more, with the development of techniques such as electrohydrodynamics and microfluidics, and a growing understanding of the self-assembly mechanism, including colloidal and intermolecular forces, liposome preparation is now richer and more controllable than ever [21]. All this makes liposomes a very attractive case of nanoparticles for theranostic purposes [26,27].

## 3. Theranostic Applications of Lipid Nanoparticles

The available technologies of LNPs for theranostic applications can be classified based on different approaches: (i) based on the types of biomaterials that are employed for their development, including lipids or in cases of hybrid systems, other molecules, e.g.,

polymers, peptides, antibodies, ligands and carbohydrates [4,8], (ii) based on their structure, for example, SLNs, NLCs LNCs and CLNs [6,7], (iii) based on their functionality and mechanism of action, which is directly linked with the properties of utilized biomaterials and final self-assembled nanostructure, e.g., stimuli-responsive behavior [4] and (iv) based on their incorporated or encapsulated diagnostic and therapeutic molecules, associated with the intended application and diagnostic modality [3]. In this section, we classify several LNP and hybrid technologies based on their non-lipid biomaterials and functionality, with emphasis on the recent trends in nanotheranostics, and some of the most recent examples in the bibliography are presented. Many of these technologies have also been combined; however, in each chapter, we emphasize specific technologies and traits of the nanoparticles.

### 3.1. Quantum Dots

Quantum dots (QDs) are semiconductor nanomaterials of size ranging between 1 and 10 nm that possess optical and electronic properties different from those of larger particles of bulk material, due to quantum mechanics. These size and composition-dependent properties render QDs very promising as materials in many application areas, including electronics, luminescence, catalysis and disease diagnostics [28]. QDs offer several advantages over other categories of fluorescent molecules. One of them is their broad absorption spectrum, combined with a narrow emission spectrum, enabling fluorescent multiplexed analysis and diagnosis. However, their lack of biocompatibility and potential toxicity limits their use in humans. An answer to this challenge is their combination with lipidic nanoparticles, for the development of functional systems for diagnostic or theranostic purposes [29].

Lopes et al., developed a hydroxyapatite-coated liposomal platform incorporating bupivacaine as a local anesthetic agent and CdSe QDs as a bioimaging agent. The formation of the hydroxyapatite coating was achieved by utilizing zwitterionic and negative phospholipids for preparing the liposomes and then adding calcium and phosphate ions to the suspension. The system would facilitate targeted delivery of the drug in situ, due to the HAP coating, as well as monitoring of in vivo drug distribution [30].

Multifunctional theranostic platforms can also integrate further modalities, such as active targeting properties to target tissues and cells. In the case of Demir et al., curcumin and carbon dot (CD)-loaded liposomes were surface-functionalized with anti-CD44 antibodies. CDs belong to the class of QDs, but have been reported to be more biocompatible, less toxic, photostable, and adaptable compared to them [31]. The final multifunctional liposomes exhibited superior effect on tumor cells compared to curcumin-loaded and curcumin/CD-loaded ones. Moreover, 3D holographic microscopy was used as an imaging tool to monitor the effect of the nanoparticles. The results were encouraging in further utilizing this theranostic platform for cancer therapy, equipped with real-time diagnosis.

Seleci et al., developed liposomes with membrane-incorporated CdSe/ZnS QDs and core-encapsulated topotecan. According to physicochemical, fluorescence microscopy and flow cytometry studies, the nanoparticles were effective in co-delivering the diagnostic and therapeutic molecules, as was evident from cellular uptake and distribution results [32]. In another study, Olerile and colleagues developed an NLC, loaded with CdTe/CdS/ZnS QDs and paclitaxel (PTX), as a parenteral theranostic DDS for cancer [33]. The nanoparticles were prepared by emulsion-evaporation and low temperature-solidification. The results from in vivo and ex vivo experiments suggested the targeted delivery of this nanosystem, with successful imaging and suppression of the tumor.

### 3.2. Inorganic Nanoparticles

LNP technology can be combined with inorganic nanoparticles that enable diagnosis and treatment. The resultant hybrid technologies are of great interest, since they combine the advantages of both classes of these nanoparticles, but have also helped in overcoming several of their individual limitations, such as solubility and toxicity issues. In addition, LNPs offer versatile solutions for the delivery of inorganic materials, where the latter can

be encapsulated inside the aqueous pore, incorporated inside the membrane or attached on the surface, depending on the desired functionality and effect. Superparamagnetic iron oxide nanoparticles (SPIONs), gold, silver and palladium are some of the candidates for this technology [7].

Saesso et al., developed liposomes incorporating SPIONs and functionalized with anti-CD20, for crossing of the blood–brain barrier and active targeting in brain lymphoma [34]. Vibrating sample magnetometry confirmed the superparamagnetic properties of the nanosystem, while cellular uptake and apoptosis were achieved in B-lymphoma cells. Finally, delivery and anticancer effect in a BBB model confirmed the potential of these nanocarriers as theranostic tools.

Wereszczyńska and Zalewski combined fatty acid derivatives of gadolinium 3+ (Gd(III)) and zinc phthalocyanine (ZnPc) in a single liposomal formulation, with the purpose of concurrent diagnosis and treatment of cancer (Figure 2) [35]. ZnPc acts as a photosensitizer for PDT and Gd(III) is an MRI contrast agent. The Gd(III) derivatives would self-assemble alongside the phospholipids, leading to orientation of Gd(III) towards the polar region, while ZnPc was incorporated inside the membrane, leading to relaxivity enhancement that might facilitate the reduction in Gd(III) dose and related toxic effects. This phenomenon is considered crucial for future development of efficient and safe MRI theranostics and is associated with the liposomal membrane structure and dynamics, altered after the incorporation of other biomaterials [36].

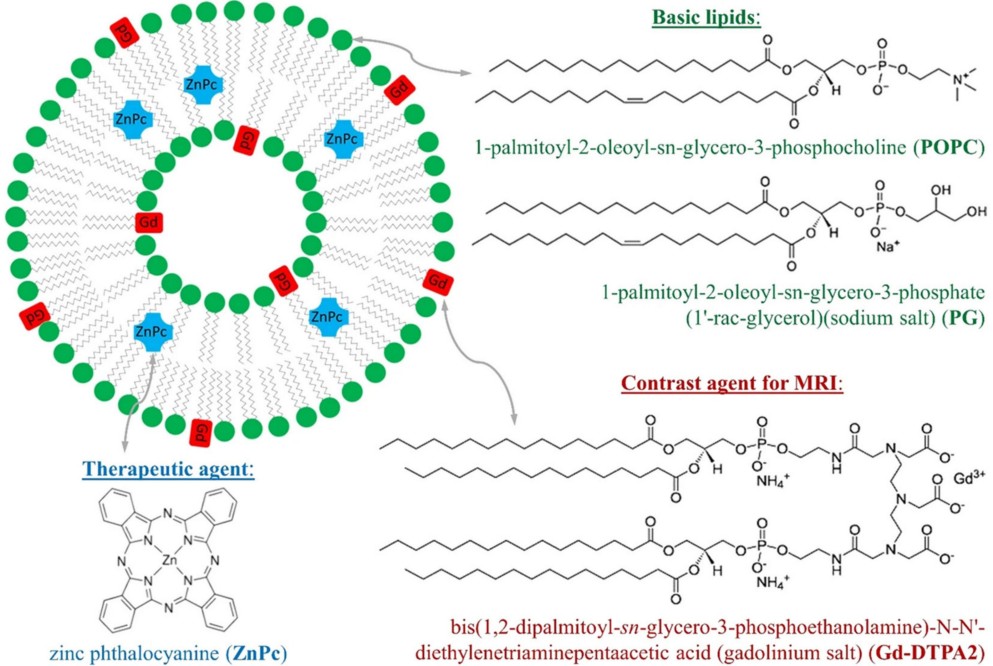

**Figure 2.** Gadolinium 3+ (Gd(III)) and zinc phthalocyanine (ZnPc)-loaded liposomes. Reproduced from [35].

Gold nanoparticles are particles with unique properties that are utilized in various imaging, diagnostic and therapeutic applications, such as CT and surface-enhanced Raman Spectroscopy (SERS). These nanoparticles can be synthesized in a tailored, precise and reproducible manner, having spheric, cubic, rod-like, cage-like or other forms. In addition, their stability, safety and ease of modification are some of the attributes that render them appropriate for diagnosis and therapy [37]. Sonkar et al., designed and developed theranostic liposomes, incorporating docetaxel (DCX) and glutathione-reduced gold nanoparticles (AuGSH) and decorated with transferrin for active targeting of the receptor in in vitro and in vivo glioma models [38]. The result was the delivery of higher amounts of DCX and

AuGSH to the brain and improved $AUC_{(0-4\,h)}$ values, compared to Docel™, while this platform was proposed as a promising nanotheranostic tool.

Another promising technology that has been integrated with LNPs, is metal–ligand coordination nanosystems, which are formed through interactions between metal ions and organic ligands. In cancer theranostics, MOFs have shown potential, as they can combine modalities of both metal ions, such as $Fe^{3+}$, as well as organic ligands, such as organic dyes. This enables the concurrent imaging by MRI and fluorescence. Lin et al. developed an indocyanine green (ICG)-$Fe^{3+}$ coordination system for US-assisted theranostics of cancer, aiming to bypass the hurdles associated with sonodynamic therapy [39]. The complex was encapsulated inside a lipid layer, leading to a multi-level self-assembling supramolecular system, using a single-step multi-level approach. The complex was administered to hepatocellular carcinoma orthotopic mouse models, increasing ROS generation and the in situ conversion of the microbubbles to nanoparticles, promoting accumulation at the target site. Superior loading efficiency and bioavailability make this nanosystem promising for non-invasive multimodal imaging.

### 3.3. Photodynamic and Photothermal Therapy

PDT utilizes photosensitizer (PS) molecules for the elimination of tumor cells. After light irritation, PSs become excited and then through one of two processes (type I or II), produce radicals and reactive oxygen species (ROS) or singlet oxygen ($^1O_2$) from triplet oxygen ($^3O_2$), thus oxidizing cellular components [40]. Their overall physiological effect includes cell ablation, inflammatory and immune responses, as well as vascular damage. Though it offers many important advantages, including almost absent invasiveness, low toxicity and low drug resistance, the application of PDT in the clinical setting is still limited, owing to the lipophilicity, short half-life and lack in tissue specificity and targeting [41]. In addition, despite the drug resistance absence, some tumor cells are able to enhance their tolerance to ROS by upregulating glutathione expression [42]. LNPs constitute a rational approach in addressing the limitations of PDT, by solubilizing and stabilizing PS molecules, by offering targeted and modified delivery of the cargo, avoiding delivery to normal tissues and related toxicity, by co-loading with other diagnostic and therapeutic agents, and by protecting PSs from exposure to blood and immune components that might lead to their degradation [40].

On the other hand, PTT is a non-invasive type of treatment that employs photothermal agents, such as metal nanoparticles, inorganic nanomaterials and small molecular organic dyes which generate hyperthermia under near-infrared (NIR) laser irradiation and ablate cancer cells [43,44]. However, hyperthermia causes cells necrosis and inflammation, leading to therapy complications, as well as injuries and serious pains to patients. This issue is traditionally addressed by co-administrating nonsteroidal anti-inflammatory drugs (NSAID) [45]. LNPs can offer better solutions to this problem.

Skupin-Mrugalska et al., demonstrated that microfluidics can be a method of choice for the one-step production of theranostic liposomes with a high entrapment degree of therapeutic and diagnostic molecules [46]. In their case, an MRI agent and a PS were co-delivered to carcinoma cell lines. In another approach on PDT, Giurguis et al. developed tunable NIR-activable liposomes that contained lipid conjugates of a benzoporphyrin derivative or IRDye 700DX, with emphasis on the role of membrane composition in the functionality of these platforms [47]. They concluded that lipid conjugates of PSs can affect the outcome of PDT based not only on their photoresponsive behavior, but also on their chemical nature and conformation inside the membrane.

Panikar et al., enhanced the PDT potential of methylene blue (MB) by attaching it on NaYF4:Yb, Er nanoparticles (UCNPs) and subsequently encapsulating the complexes inside liposomes [48]. The final formulation exhibited enhanced ROS generation, while active targeting properties were achieved by membrane incorporation of a lipid derivative of their newly developed anti-HER-2 peptide (Figure 3). In another recent study, hypericin, a molecule for PDT and photodiagnosis, was incorporated in three different types of nanove-

hicles, based on the copolymer F127 and dipalmitoyl-sn-3-glycerol-phosphatidylcholine (DPPC) [49]. The aim was to improve its photophysical properties, by preserving its monomeric form. From the available platforms, the mixed copolymer–lipid system proved superior in most aspects.

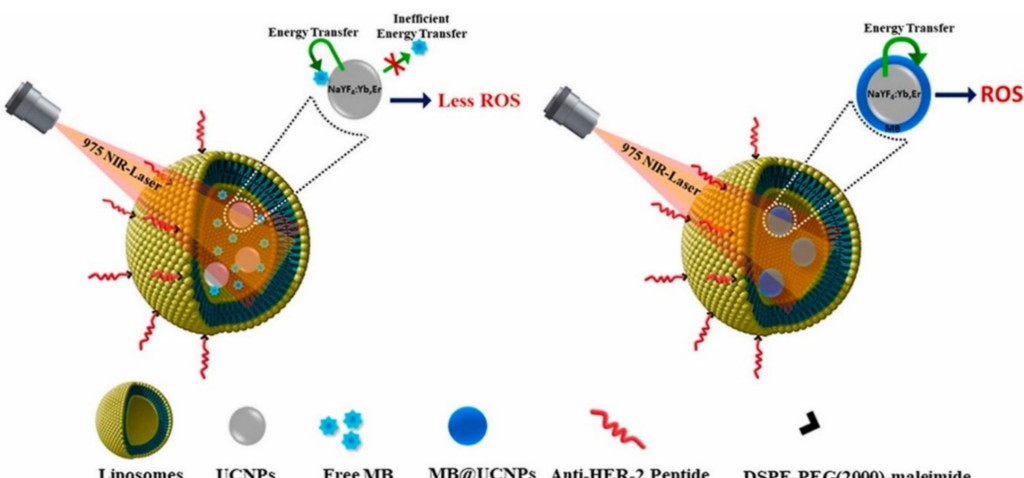

**Figure 3.** Active targeting liposome encapsulated methylene blue (MB)-NaYF4:Yb, Er nanoparticles (UCNPs) are activated by NIR laser to produce ROS. On the left, MB is free, while on the right, it is attached on the UCNPs, leading to greater ROS generation. Reprinted with permission from Ref. [48]. Copyright © 2021, Elsevier.

Another promising candidate for PDT and PTT is porphyrin-lipid nanoparticles or porphysomes, which are vesicles composed of pyropheophorbide-conjugated phospholipids [50]. These nanoparticles are promising theranostic agents for a wide range of applications, including drug delivery, phototherapy, magnetic resonance imaging (MRI) and PET. They end up inside tumors due to the enhanced permeability and retention (EPR) effect and after their entrance into the cancer cells, they are disrupted into pyrolipid subunits. Both the intact and the disrupted state can be exploited for therapeutic purposes.

Previous studies would primarily focus on the use of porphysomes as PTT agents, suggesting that they can be utilized for PDT only though target-triggered activation. Guidolin et al. showed through animal studies that porphysomes are effective in PDT without any modification, by comparing them with Photofrin®, a porphyrin-based PDT agent that is clinically approved [51]. They concluded that multimodal diagnostic and therapeutic applications arise from the intrinsic structure of porphysomes and that they also enable concurrent PDT and PTT. What is more, skin photosensitivity, which is a limiting factor in PDT, may be significantly lower with porphysomes than with Photofrin®.

### 3.4. Thermosensitive Liposomes

Lysolipid-containing temperature-sensitive liposomes (LTSLs) are a promising endeavor in the area of drug delivery. They undergo gel-to-liquid crystalline phase transition under conditions of hyperthermia (HT), releasing their cargo in a spatiotemporal way [52]. In the case of ThermoDox®, this hyperthermia (40–42 °C) is induced by applying local radiofrequency ablation and high focused intensity ultrasound (HIFU) [53]. The product has undergone a Clinical Phase 3 for the treatment of hepatocellular carcinoma and is currently under Clinical Phase 1 for liver cancer. In fact, local HT has been employed in chemotherapy to enhance drug accumulation through tissue perfusion, as well as to sensitize tumor cells to treatment. With LTSLs, it is exploited to increase drug bioavailability and decreasing peripheral toxicity [54,55].

LTSLs can be combined with SPIONs or other metallic nanoparticles on a single hybrid nanoplatform [56–58]. In this way, thermo- and magneto-responsive behaviors can be integrated to offer multifunctionality. This is achieved by utilizing the classic phos-

pholipid DPPC, the lysolipid 1-stearoyl-2-hydroxy-sn-glycero-3-phosphocholine (MSPC) that induces membrane pore formation during the thermoresponsive phase transition, and a component to promote interfacial stabilization and biological stability, such as PEG-lipid 1,2-distearoyl-sn-glycero-3-phosphoethanolamine-N-[methoxy(polyethyleneglycol)-2000] (DSPE-PEG2000) [59]. In addition, SPIONs, gold or copper sulfide nanoparticles, when incorporated inside the liposomal membrane, offer bioimaging. As a result, these lysolipid-containing temperature-sensitive magnetoliposomes (mLTSLs) initially act as MRI contrast agents and when they reach the target tissue, they are destabilized by hyperthermia, which originates from external sources or from the magnetic nanoparticles that are stimulated [60,61].

At the same time, local HT and the use of specific drug molecules, such as doxorubicin (DOX), have been associated with upregulation of cell death ligand 1 (PD-L1), which create an immunosuppressive environment inside tumor tissues [62,63]. Based on this, Ma et al. studied the beneficial effect of cell death protein 1 (PD1) blockade, in combination with photothermally activated mLTSLs with encapsulated DOX [64]. Drug delivery and MRI were achieved, while tumor growth was significantly reduced by coadministration of anti-PD1 monoclonal antibodies (mAb). In this case, the drug was delivered inside the liposomal core, while nitrodopamine palmitate-coated iron oxide nanoparticles (IO NPs) were incorporated inside the membrane.

One of the main challenges in the development of mixed/chimeric advanced drug delivery nanosystems (aDDnSs) is to have a well-defined, robust and scalable manufacturing process. The nanoscale is one factor that renders formulation difficult, while another is the complexity of the system. Cheung and colleagues managed to develop a SPION-incorporating LTSL through a scalable nanoprecipitation method, with high reproducibility and stability [59]. These nanoparticles offer multimodal imaging and targeted drug delivery, which are controlled through local hyperthermia and the SPION behavior, modulated, e.g., by a magnetic field or NIR irradiation.

*3.5. Cell Membranes*

The utilization of cell membranes as a tool to effectively deliver bioactive agents is the next step in terms of biocompatibility and biological stability. The cell membrane is essential in fundamental cellular functions, including compartmentalization, self-identification, bio-interfacing and signal transduction [65]. Cell membrane-based nanoparticles (CMBNs) are cell-material nanohybrids that combine the advantages of biomimicry and bio-functionality, being able to mimic and interact with the complex biological microenvironment [66]. Inside this technology, there are both synthetic and natural elements, where the latter are proteolipid vesicles that act as a "trojan horse", camouflaging nanomaterials that carry drug molecules, genes or imaging agents. CMBNs are a new class of DDSs that integrate the unique biomimetic and functional properties of cell membranes and the engineering versatility of synthetic nanomaterials [67].

Depending on the intended application and target site, the membranes can originate from cancer cells, red or white blood cells, platelets, mesenchymal stem cells or neutrophils, with each class having its own advantages. For instance, cancer cell coating offers increased circulation and tumor affinity and homing, facilitating theranostics, while reducing any potential side effects [65,68]. The main mechanism that drives these particles is the affinity of the utilized membrane for the relevant tissue. One very important advantage of this new class of nanoparticles is that they can escape the immune system and have prolonged circulation time, which is a prerequisite for the EPR effect and tumor targeting [69].

Rao et al., prepared erythrocyte membrane-coated magnetic nanoparticles by microfluidic electroporation, instead of conventional extrusion [70]. The formulation was tested in vivo on MCF-7 human breast tumor xenografts and exhibited superior effect, apparently due to the achievement of better coating results. In particular, $Fe_3O_4$ nanoparticles and red blood cells were infused inside a microfluidic device, where electric pulses promoted the entry of the former inside the latter. The resultant nanoparticles exhibited superior

magnetic and photothermal properties that were utilized for MRI and PTT in mice. Furthermore, serving scale-up and industrialization, the authors claim that the method offers accurate control of the size and function of the vesicles that are used for coating. At the same time, autologous extraction of RBCs, combined with a convenient production method, might facilitate personalized theranostics that will be compatible with each individual's immune system.

Neutrophils are another promising approach in cell membrane-mediated drug delivery and find extensive application in glioma treatment, since they can penetrate the blood brain tumor barrier (BBTB) and are the most abundant immune cell class. In this context, Xue and co-workers developed neutrophils with encapsulated PTX-loaded liposomes for post-operative glioma recurrence [71]. The stimulus for the release of PTX from the neutrophils was the high concentration of inflammatory signals, leading to slower tumor recurrence and growth, as well as improved survival, even though the tumor regeneration was not prevented. Such studies pave the way for clinical exploitation of physiologically derived immune cells as DDSs.

The selective deprivation of nutrient supply and metabolic pathways of tumor cells can improve anticancer treatment. Li et al. designed and developed a cascade bioreactor that enabled concurrent starvation of cancer cells and PDT [72]. They achieved this by incorporating glucose oxidase and catalase molecules inside a porphyrin MOF of a porous coordination network, which was then camouflaged with a cancer cell membrane. Cancer cell targeting and tumor retention were enhanced, owing to the biomimetic camouflage, immune escape and homotypic targeting. From there on, cancer cell internalization was followed by the bioreactor promoting microenvironmental oxygenation, leading to decomposition of intracellular glucose and enhancement of cytotoxic singlet oxygen ($^1O_2$) under light irradiation. The results were synergistic long-term cancer starvation and robust PDT, effectively inhibiting tumor growth after a single administration. In similar context, Chen and coworkers developed poly lactic-co-glycolic acid (PLGA) nanoparticles that incorporated ICG in the polymeric core and were coated with cancer cell membrane as a surrounding shell [73]. The nanoparticles exhibited homologous targeting, good photothermal properties and excellent imaging properties, based on fluorescence and photoacoustics (Figure 4). The membrane shell facilitated cell endocytosis, homologous targeting and in vivo tumor accumulation, while liver and kidney uptake were decreased. Finally, the nanosystems exhibited very efficient PT and elimination of the xenograft tumor, suggesting that cancer cell membranes are very promising as biomimetic components of DDSs for cancer imaging and therapy.

### 3.6. Lipoprotein Nanoparticles

Lipoproteins have the endogenous role of transferring hydrophobic molecules, such as cholesterol and other lipids, between different sites inside the organisms. Recently, the understanding of their function and role, alongside their biocompatibility and biomimicry, have led to the development of lipoprotein-inspired nanosystems, which are promising as DDSs for therapeutic, diagnostic and theranostic purposes [74]. Such nanosystems can circumvent the reticuloendothelial system, to have a prolonged circulation time, while their very small size allows them to efficiently penetrate tumors. In addition, lipoproteins have high affinity for endogenous receptors that can be found on cancer cells. Their structure includes a hydrophobic core, surrounded by a phospholipid monolayer that contains unesterified cholesterol and apolipoproteins [4].

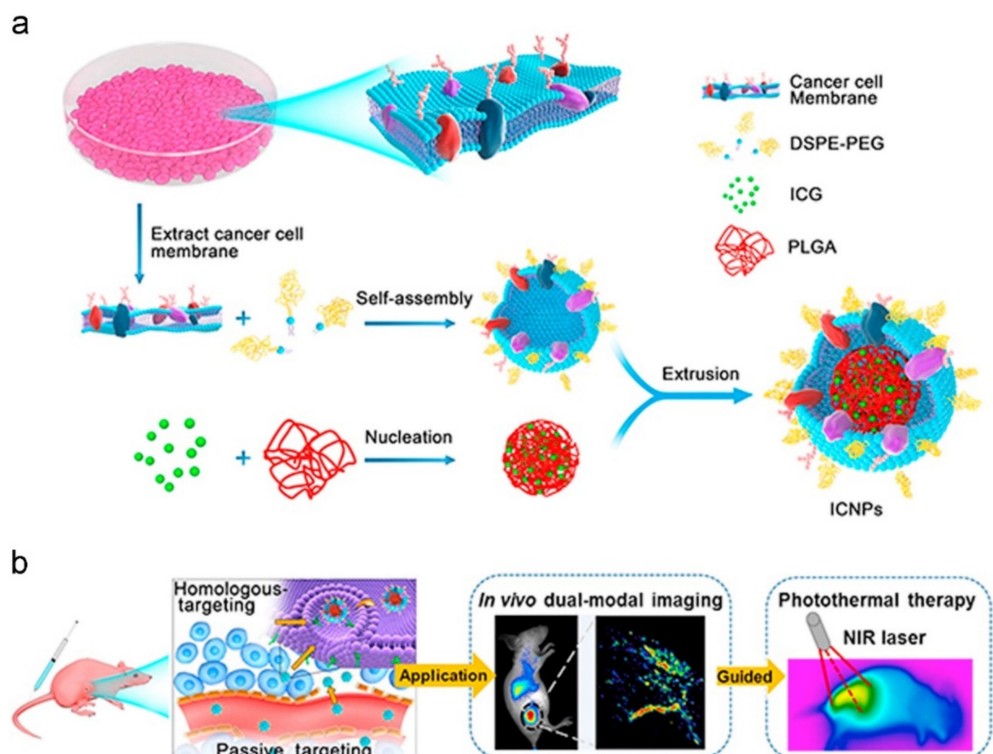

**Figure 4.** Biomimetic cancer cell membrane-coated nanoparticles for targeting of homologous cancer cells, for dual-modal imaging and PTT. (**a**) Preparation procedure of ICNPs. (**b**) Schematic of homologous-targeting ICNPs for dual-modal imaging guided photothermal therapy. Reprinted with permission from Ref. [73]. Copyright © 2016, ACS Nano.

Sheng et al., developed a nanoplatform, based on high-density lipoproteins (HDLs), in which they incorporated ICG for NIR-activated fluorescence imaging, PTT and PDT [75]. In addition, they decorated the nanoparticles with a tumor-homing iRGD peptide via conjugation, for active targeting. The peptide was attached by cross-linking, which led to the organization of a scaffold that could maintain the nanoparticle structure. The HDL-like nanoparticles could penetrate tumors, facilitating effective PTT and PDT under NIR light irradiation. At the same time, He and colleagues built a nanotheranostic system based on an HDL-mimicking peptide–phospholipid scaffold (HPPS), with hTfR (human TfR) monoclonal antibody (mAb) decoration, complexed siRNA, and the fluorophore DiR-BOA incorporated inside the core [76]. DiR-BOA acts as an imaging tool, while the nanoparticles could efficiently target and cause HEPG2 apoptosis.

## 4. Clinical Translation of Lipid Nanoparticles for Theranostics

Even though the first FDA-approved nanomedicine was a liposomal formulation, even though extensive research has been conducted on lipidic nanoparticles as DDSs and theranostic vehicles and despite their distinctive advantages, no nanotheranostic lipid-based formulation has yet been approved (Table 2). Lipids belong to the GRAS class, are generally biocompatible, show minimal toxicity and have been so far utilized to alleviate the heavy toxic profile of certain chemotherapeutic molecules; however, the need for expedient information on their case-by-case suitability, including safety and effectiveness, still exists [8,20].

**Table 2.** Lipid nanoparticles currently under clinical evaluation.

| Nanosystem | Title | Type of Cancer | Sponsor/Agency | Clinical Trial ID | Phase |
|---|---|---|---|---|---|
| Liposome | Phase IIb Study Evaluating Immunogenic Chemotherapy Combined With Ipilimumab and Nivolumab in Breast Cancer (ICON) | Breast Cancer | Oslo University Hospital | NCT03409198 | 2 |
| Liposome | To evaluate 188Re-BMEDA-liposome in Patient With Primary Solid Tumor in Advanced or Metastatic Stage | Tumors | Institute of Nuclear Energy Research, Taiwan | NCT02271516 | 1 |
| Liposome | EphA2 siRNA in Treating Patients With Advanced or Recurrent Solid Tumors | Solid Tumors | M.D. Anderson Cancer Center | NCT01591356 | 1 |
| Liposome | Targeted Chemotherapy Using Focused Ultrasound for Liver Tumours (TARDOX) | Liver Tumors | University of Oxford | NCT02181075 | 1 |

The prerequisites for clinical translation regard the nanocarrier itself, including small size (equal to or below 100 nm), appropriate morphology and low surface charge, its interactions with the bioactive molecules, requiring satisfactory incorporation/encapsulation efficiency and stability, but also the interactions between the loaded nanocarrier and the biological system. Protein binding, biological stability, biodistribution, potential toxicity, immunogenic reactions, metabolism and clearance are all of extreme concern when developing such nanosystems. Additionally, the nanoparticulate systems need to be scalable and producible in a robust and reproducible manner [77]. This task is difficult, especially in cases where precise chemistry is required and multifunctional units exist on the nanosystem. Since these are complex systems, a holistic approach that employs a broadrange arsenal of analysis tools is necessary to predict and control their properties. Indeed, lipid-based nanoparticles and specifically liposomes have been proven to be chaotic deterministic in nature, with their fundamental properties depending on various controllable and non-controllable factors [78]. This means that new tools are required to evaluate their self-assembled properties and biological behavior, including biophysical and thermodynamic approaches [11]. Finally, apart from the great bench-to-clinic gap, there is also a great gap between science and regulation. The lack of clear and specific guidelines on the nanomedicines of each class, as well as their follow-on nanosimilar products, limits the translation of these products to the market [79,80].

The solutions to these challenges orientate towards the production, biological and clinical parts. First of all, the gaps between the academia, the pharmaceutical industry, the clinic and the regulatory authorities must be bridged in the near future. Strong collaboration between these entities will greatly benefit the field. For example, the academia can assist the industry in the scale-up and good manufacturing process of innovative nanoformulations, since these do not fall in the same category with classical pharmaceuticals and their production presents newfound challenges [7]. In addition, preclinical studies with implemented standardized nanotoxicology protocols can offer invaluable information about the cytotoxicity, immunotoxicity and genotoxicity of nanoparticles and testing on multiple animal models is essential for clinical translation and evaluation of potential future patient risk [81]. In order to design, develop and optimize nanoformulations based on their potential interactions with biological components and barriers, new software and artificial intelligence (AI) tools can be applied on both the research and production levels [82]. What is more, the synergistic combination of nanomedicine and AI brings us one step closer to precision medicine, breaking down patient and disease heterogeneities and probably solving a critical aspect of clinical testing [83].

## 5. Conclusions

The advance in the research of LNP technologies and applications is tremendous, providing more and more complex nanosystems that present multifunctional properties. Additionally, the current COVID-19 situation has attracted much attention to these nanoparticles, with the industry now having much more experience in producing and evaluating them. However, there still seems to be some way until nanotheranostic DDSs and especially aDDnSs are fully understood and approved for human use. The reason is of course their limited translation to the clinic, as well as the limited presence of regulatory guidance on their safety and efficacy standards. Regarding the first point, there are not many studies that demonstrate how imaging modalities can be utilized to alter and improve therapeutic plans and disease outcomes. If achieved, this approach will pave a new road to personalized medicine. LNPs are the most well-studied class of nanoparticles and since they are bio-inspired, biocompatible and versatile, they are the a priori safe choice for developing nanotheranostic tools.

**Author Contributions:** Conceptualization, N.N.; writing—original draft preparation, N.N. and C.D.; writing—review and editing, N.N. and C.D. All authors have read and agreed to the published version of the manuscript.

**Funding:** This research received no external funding.

**Conflicts of Interest:** The authors declare no conflict of interest.

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
