# Peer review of "Lipid Nanoparticles as Platforms for Theranostic Purposes: Recent Advances in the Field"

_jnt, doi:10.3390/jnt3020006_

Round 1

Reviewer 1 Report

The manuscript entitled “Lipid Nanoparticles as Platforms for Theranostic Purposes: Recent Advances in the Field” is intended to be a review dedicated to complex systems based on lipid carriers for diagnosis and treatment of different illnesses.

My suggestions/remarks are listed below:

  • Minor English mistakes, such as line 38 “However, the to fields need to be developed and applied synergistically, especially in the case of complex diseases, in order to ensure their maximum efficiency” - this sentence should be reformulated, it is unclear.
  • The manuscript is well written and the sections are well organized.
  • Not all figures have copyright permissions

I propose the manuscript to be published after minor modifications!

Author Response

The manuscript entitled “Lipid Nanoparticles as Platforms for Theranostic Purposes: Recent Advances in the Field” is intended to be a review dedicated to complex systems based on lipid carriers for diagnosis and treatment of different illnesses.

Comment 1:

Minor English mistakes, such as line 38 “However, the to fields need to be developed and applied synergistically, especially in the case of complex diseases, in order to ensure their maximum efficiency” - this sentence should be reformulated, it is unclear.

Answer 1:

We have checked the entire manuscript and corrected any language mistakes, according to the reviewer’s comment.

Comment 2:

The manuscript is well written and the sections are well organized.

Answer 2:

We thank the reviewer for the comment.

Comment 3:

Not all figures have copyright permissions.

Answer 3:

Figures 1 and 2 are from open access journals that do not require permission. Figure 3 has been replaced and permissions have been obtained.

Comment 4:

I propose the manuscript to be published after minor modifications!

Answer 4:

We sincerely thank the reviewer for his/her comments and this opportunity.

Reviewer 2 Report

Manuscript covers a loot of new content in lipid Nps. I recommend authors to includes latest developments in vaccines preparations using lipid nps. 

Author Response

Manuscript covers a loot of new content in lipid Nps.

Comment:

I recommend authors to includes latest developments in vaccines preparations using lipid nps.

Answer:

We recognize the significance of LNPs in vaccine development and we have added extra elements in the second section of our review. However, vaccine formulations being far from the theranostics concept, we would prefer not to extend our reference to this topic.

Reviewer 3 Report

The review is focused on recent literature data on lipid nanoparticles (LNPs) that are generally considered as the most important nanocarriers for drug delivery in medicine. In fact these biomaterials are actually a resource for fighting pandemia. The paper present a very detailed description and accurate analysis of biomaterials employed for LNPs formulation and their recent application in the field of nanotheranostics

Author Response

The review is focused on recent literature data on lipid nanoparticles (LNPs) that are generally considered as the most important nanocarriers for drug delivery in medicine. In fact these biomaterials are actually a resource for fighting pandemia. The paper present a very detailed description and accurate analysis of biomaterials employed for LNPs formulation and their recent application in the field of nanotheranostics.

Answer:

We would like to thank the reviewer for his comments and appreciation of our work.

Reviewer 4 Report

The review article entitled “Lipid Nanoparticles as Platforms for Theranostic Purposes: Recent Advances in the Field” summarizes the general and theranostic application and the clinical translation of lipid nanoparticles. The article is well written, provides useful information for the reader regarding to the application possibilities of LNPs, I have only few comments which should be clarified, before further consideration.

Niosomes are a nonionic surfactant-based vesicles that usually contains a single hydrophobic tail, however in Fig. 1. two tail phospholipids were presented as wall-forming components. Please modify the image.

It would be worth to collect the most commonly applied functionalized phospholipids in a table, which can be applied for constructing advanced LNPs.

The author indicate LNPs can be exploited for the development of recent COVID-19 vaccines, however, no specific details are presented.  What kind of lipids can be applied? Which LNPs are the most suitable for that purpose? How can LNPs used for mRNA delivery?

Author Response

The review article entitled “Lipid Nanoparticles as Platforms for Theranostic Purposes: Recent Advances in the Field” summarizes the general and theranostic application and the clinical translation of lipid nanoparticles.

Comment 1:

The article is well written, provides useful information for the reader regarding to the application possibilities of LNPs, I have only few comments which should be clarified, before further consideration.

Answer 1:

We deeply thank the reviewer for his/her constructive feedback.

Comment 2:

Niosomes are a nonionic surfactant-based vesicles that usually contains a single hydrophobic tail, however in Fig. 1. two tail phospholipids were presented as wall-forming components. Please modify the image.

Answer 2:

We would like to thank the reviewer for bringing this issue to our attention. We have modified Figure 1 accordingly.

Comment 3:

It would be worth to collect the most commonly applied functionalized phospholipids in a table, which can be applied for constructing advanced LNPs.

Answer 3:

We thank the reviewer for the suggestion. We have now included Table 1, in which some of the most common chemically modified and functionalized phospholipids are presented.

Comment 4:

The author indicate LNPs can be exploited for the development of recent COVID-19 vaccines, however, no specific details are presented. What kind of lipids can be applied? Which LNPs are the most suitable for that purpose? How can LNPs used for mRNA delivery?

Answer 4:

We have added extra elements on LNP vaccine formulations in the second section of our review, according to the reviewer’s comment.

Round 2

Reviewer 4 Report

The authors adressed all my concerns.